# The Portuguese version of the self-report form of the DSM-5 Level of Personality Functioning Scale (LPFS-SR) in a community and clinical sample

Rute Pires[1,2]*, Joana Henriques-Calado[1,2], Ana Sousa Ferreira[1,3], João Gama Marques[4,5], Ana Ribeiro Moreira[6], Bernardo C. Barata[7], Marco Paulino[1,4], Leslie Morey[8], Bruno Gonçalves[1,2]

1 Faculdade de Psicologia, Universidade de Lisboa, Alameda da Universidade, Lisboa, Portugal, 2 CICPSI, Faculdade de Psicologia, Universidade de Lisboa, Alameda da Universidade, Lisboa, Portugal, 3 Instituto Universitário de Lisboa—Business Research Unit (BRU-IUL), Lisboa, Portugal, 4 Clínica Universitária de Psiquiatra e Psicologia Médica, Faculdade de Medicina, Universidade de Lisboa, Lisboa, Portugal, 5 Consulta de Esquizofrenia Resistente, Hospital Júlio de Matos, Centro Hospitalar Psiquiátrico de Lisboa, Lisboa, Portugal, 6 Centro Hospitalar de Lisboa Ocidental, Hospital de Egas Moniz, Lisboa, Portugal, 7 Departamento de Psiquiatria e Saúde Mental, Centro Hospitalar Barreiro Montijo, Barreiro, Portugal, 8 Texas A&M University: College Station, TX, United States of America

* rpires@psicologia.ulisboa.pt

## Abstract

The Level of Personality Functioning Scale–Self-Report (LPFS-SR) operationalizes Criterion A of the DSM-5 Alternative Model for Personality Disorders. The current study aimed 1) to examine the internal consistency of the Portuguese version of the LPFS-SR in a community sample and a clinical sample, 2) to compare non-clinical participants ($N = 282$, $M_{age} = 48.01$, $SD = 10.87$) with two samples of clinical participants, one composed of patients with a personality disorder diagnosis (PD sample, $n = 40$, $M_{age} = 46.18$, $SD = 13.59$) and the other of patients with other psychiatric diagnoses (OD sample, $n = 148$, $M_{age} = 49.49$, $SD = 11.88$), with respect to LPFS-SR dimensions and total score, 3) to examine the capacity of the LPFS-SR to discriminate between samples through the ROC curve analyses, and 4) to examine the factor structure of the Portuguese version of the LPFS-SR. The Portuguese version of the LPFS-SR revealed adequate internal consistency results, akin to the original data, in the community and clinical samples. The community sample differed significantly from both clinical samples in all the LPFS-SR dimensions and total score. The ROC curve analysis indicated an optimal cut-off for the total score of 272.00, corresponding to a sensitivity of 75% and a specificity of 89%, in the PD vs. community samples. The LPFS-SR total score discriminative capacity between the PD and OD samples was lower, albeit also significant (area-under-the-curve of .63; $p = .027$; 95% CI: .52-.74). The current study provided evidence of the LPFS-SR's unidimensionality in both community and clinical samples. Although this study has limitations, its findings contribute to a deeper understanding of the LPFS-SR construct, as well as to its cross-cultural validation.

**Data Availability Statement:** All relevant data are within the manuscript and its Supporting Information files.

**Funding:** - This work received national funding from the FCT – Fundação para a Ciência e a Tecnologia, I.P [Foundation for Science and Technology], through the Research Centre for Psychological Science of the Faculty of Psychology, University of Lisbon (UIDB/04527/2020; UIDP/04527/2020) and the Business Research Unit, BRU-IUL (UIDB/00315/2020). The funders had no role in the study design, data collection and analysis, decision to publish, or preparation of the manuscript.

**Competing interests:** The authors have declared that no competing interests exist.

# Introduction

Both the fifth edition of the Diagnostic and Statistical Manual of Mental Disorders (DSM; [1]), and the ICD-11 Classification of Personality Disorders (WHO, [2]) consider the severity of impairments in self- and interpersonal functioning as the core feature of personality disorders (PD), although personality traits are also important for describing the stylistic manifestations of personality dysfunction. In the DSM-5 Alternative Model for Personality Disorders [1], the specific pathological trait patterns (Criterion B) that characterize each PD may be assessed through the Personality Inventory for DSM-5 (PID-5; [3]). Substantial research has been conducted on this inventory over the last decade in a range of countries across Western and Eastern cultures [4–18]. Notwithstanding the centrality of personality dysfunction in the diagnosis of PD, fewer studies have focused on the level of personality functioning (Criterion A) relative to traits (Criterion B) over the last decade. This is most likely due to the fact that the DSM-5 has proposed a clinician-rated scale for its characterization (LPFS; [19]), which is less suitable for research purposes than self-report measures.

The LPFS model describes personality functioning on a continuum, differentiating five levels of impairment ranging from little or no impairment (Level 0) to extreme impairment (Level 4). Although personality functioning is a single dimension that reflects the global severity level of personality, impairments can manifest themselves in self (Identity and Self-Direction) and/or interpersonal functioning (Empathy or Intimacy). Moderate (Level 2) or greater impairment in personality functioning, manifested by problems in two or more of these dimensions (Identity, Self-Direction, Empathy, or Intimacy) is required for the diagnosis of PD [1].

Given the centrality of personality dysfunction in PD assessment and diagnosis, self-report measures for Criterion A, which are vital to furthering research, have emerged. Among other authors (see [20, 21]), Hutsebaut et al. [22–24] and Huprich et al. [25] have developed self-report measures to assess the LPFS construct, but these measures do not strictly adhere to the LPFS descriptors presented in the DSM-5. In response, Morey [26] developed an 80 item self-report measure that directly operationalizes the LPFS construct as described in the DSM-5. Unlike the previously reported measures, the items of the LPFS-SR were written to target the LPFS characteristic descriptors of impairment in self-functioning (e.g., different levels of identity integration and self-direction) and interpersonal functioning (e.g., different capacities for empathy and intimacy) presented within Table 2 of the DSM-5 AMPD [1]. This indicator-by-indicator mapping of the LPFS into self-report permits a close examination of the precise strengths and weaknesses of the description of global personality pathology provided in the AMPD. Moreover, its adequate psychometric properties allow for a deeper understanding of the ways personality dysfunction and personality traits interweave. Although the discussion surrounding the relationship between personality functioning and traits is beyond the scope of this manuscript (see [20] for a revision), it should be noted that on the 10th anniversary of the publication of the DSM-5, this issue is still under debate [20, 27]. Although conceptually distinct, there appears to be a considerable empirical overlap between maladaptive personality functioning and maladaptive personality content, which raises questions as to the maintenance of both criteria in future editions of the AMPD [20, 28–44]. Leaving behind an "either-or" discussion, Criterion A components span across the normal adaptative realm, which is not the case of all pathological traits, and from a clinical stand point, this constitutes a noticeable and useful difference between Criterion A and B. Conversely, considering that "pathological traits represent pieces of disturbed self and other narratives" [45], personality dysfunction and traits cannot be independent.

In line with the assumption that Criterion A reflects a single dimension of global personality dysfunction, research has supported the unidimensional structure of the LPFS-SR, albeit

one that can be manifested in both interpersonal and intrapsychic ways [26, 46, 47]. However, Sleep et al. [43] report that in their data neither confirmatory nor exploratory factor analyses provided evidence for a single-factor structure and called for a re-evaluation of the measure's structure. Morey [20, 48] and Sleep et al. [43, 44] have exchanged arguments regarding the dimensionality of the LPFS-SR, with Morey stating that in Sleep's study a confirmatory factor analysis model (CFA) was applied with its usual restrictions, to test a four-factor model, revealing, however, a poor fit in several indices. Additionally, this four-factor model would be incoherent with the Criterion A assumption of an underlying continuum of global personality disfunction. In turn, Sleep responded, stating that Morey's CFA analysis for a single-factor model also revealed a poor fit and drew attention to the author's Principal Component Analyses (PCA) results, which extracted a single factor, thus rendering these models somewhat unconvincing. The current study pertains to contribute to this ongoing debate, examining the factor structure of the Portuguese version of the LPFS-SR using a method that determines the optimal balance between model fit and model complexity [49].

This study focused on the psychometric properties of the Portuguese version of the LPFS-SR, seeking to contribute to its cross-cultural validity. It addressed: 1) the internal consistency of the Portuguese LPFS-SR in community and clinical samples; 2) the analysis of the community sample results on the Portuguese LPFS-SR and its comparison to those obtained by two Portuguese clinical subsamples, one composed of personality disorders (PD sample) and the other of other psychiatric diagnoses (OD sample); 3) the capacity of the LPFS-SR to discriminate between the PD sample and the community sample and between both clinical samples (PD and OD samples) through ROC curve analyses; 4) the factor structure of the Portuguese version of the LPFS-SR.

At the outset, the internal consistency of the Portuguese LPFS-SR was expected to be akin to Morey's original data [26]. Aiming to validate the Portuguese version of the LPFS-SR and its clinical usage, the clinical samples were expected to obtain higher results on the LPFS-SR than the community sample. Considering that the LPFS-SR is intended to assess core personality functions in the interest of diagnosing personality disorder, the PD sample was also expected to yield higher results on the LPFS-SR than the OD sample. Furthermore, we sought to replicate previously published studies on clinical populations (e.g., Hemmati et al. [50]) in which ROC curve analyses were used to examine the capacity of the LPFS-SR to discriminate between the PD sample and the community sample and between both clinical samples (PD and OD samples). Finally, we expected to confirm the unidimensional structure of the LPFS-SR in the community and clinical samples, which has been supported by research [26, 46, 47, 50] and is consistent with the assumption that Criterion A reflects a single dimension of global personality dysfunction.

## Method

This study was conducted under a research project entitled "Personality and Psychopathology III: Validation Studies of the Alternative DSM-5 Model for Personality Disorders in the Portuguese Population" approved by the Ethics Committee of the Faculty of Psychology, Lisbon University. Participants were recruited between 1st November 2021 and 1st July 2022. All the participants provided written informed consent.

### Sample

The community sample consisted of 282 volunteers aged between 21 and 83 years ($M_{age}$ = 48.01, $SD$ = 10.87, 46.8% male, 53.2% female), recruited from the relatives and acquaintances of undergraduate students of the University of Lisbon attending a subject whose programme

focused on personality and individual differences. Each student who volunteered to participate in the sample collection process, applied the research protocol to two adults from the community and received an award in the final mark of the subject. Participants from the general population answered the research protocol on paper, at home, with the responses returned in a sealed envelope. They did not receive any incentive to participate.

The clinical sample was collected for research purposes, after establishing partnerships with several mental health units. The sample was composed of 188 patients aged between 18 and 77 years ($M_{age}$ = 48.79, $SD$ = 12.30, 45.2% male, 54.8% female), who were undergoing treatment at the time at mental health units. In each affiliated mental health unit, a clinician coordinated the sampling procedures and selected the participants from the clinical databases of the institution, or from whom they had been referred. The collected clinical sample relied on the direct clinical evaluation of several psychiatrists, whose diagnosis had been previously discussed and agreed upon by a clinical team. Patients were selected according to their DSM-5 diagnosis (Sections I, II) and the study's inclusion and exclusion criteria. The study's inclusion criteria were individuals aged 18 years or above. Diagnoses of intellectual disability, schizophrenia, and major and mild neurocognitive disorders were the exclusion criteria. Some patients answered the research protocol during their short- term hospitalizations, others were outpatients, admitted sequentially in the sample whenever they had a follow-up consultation. This clinical sample was divided into two clinical subsamples, one composed of patients diagnosed with a personality disorder ($n$ = 40, $M_{age}$ = 46.18, $SD$ = 13.59, 47.5% male, 52.5% female), and the other of patients with several psychiatric diagnoses, other than personality disorders ($n$ = 148, $M_{age}$ = 49.49, $SD$ = 11.88, 44.6% male, 55.4% female). Regarding the personality disorders subsample, patients were assigned to this group whenever they had a personality disorder diagnosis as a major diagnosis or in comorbidity. Comorbid diagnoses included depressive disorders, bipolar and related disorders, and anxiety disorders. As for the other diagnoses group, the major diagnoses comprised depressive disorders, bipolar and related disorders, anxiety disorders, obsessive-compulsive and related disorders, substance-related and addictive disorders and trauma and stressor related disorders. Comorbid diagnoses included depressive disorders, substance-related and addictive disorders, trauma and stressor related disorders, anxiety disorders, and obsessive-compulsive and related disorders.

## Instruments

**Level of Personality Functioning Scale–Self-Report (LPFS-SR; [26]).** The LPFS-SR is a self-report measure that operationalizes Criterion A of the DSM-5 Alternative Model for Personality Disorders. It consists of 80 items, rated on a 4-point Likert scale ranging from 1 (totally false, not at all true) to 4 (very true) which characterize a global severity level and four dimensions of personality dysfunction (Identity, Self-Direction, Empathy, and Intimacy). The scoring of the LPFS-SR is designed so that higher scores represent personality problems of increasing severity.

The authorization to translate the LPFS-SR into Portuguese was obtained from the author of the test. Upon authorization, an expert in the field of personality research and proficient in the English language translated the original English items into Portuguese. The Portuguese translation of the LPFS-SR was independently evaluated by two senior personality researchers, all well acquainted with test development procedures. The final wording was obtained after consensus among the two researchers and the translator. This preliminary Portuguese version of the test was then given to a native English speaker who is also a professional translator for back-translation and was finally sent to the author of the LPFS-SR for revision and approval.

## Data analysis

Analyses were undertaken with the IBM SPSS Statistics (Version 27) and FACTOR software (version 12.04.23) [51]. The FACTOR software freely obtained from Rovira i Virgili University, is one of the most complete statistical programs for conducting an EFA, given that it congregates both CFA indexes and EFA. FACTOR includes the most current criteria for fundamental decisions: the correlation matrix to be used, the method to be adopted for extracting common factors, the number of factors to be retained and the rotation method [52]. Additionally, this software provides some goodness-of-fit indexes and the corresponding confidence intervals are based on bootstrap techniques, which are unusual in EFA algorithms.

Descriptive statistics for the LPFS-SR dimensions and total score were obtained, as well as the item-total correlations and the inter-scale correlations in the community and clinical samples. The Cohen's d was used as a measure of effect size, to study the mean score differences between the Portuguese community sample and the US original sample. The effect size was considered small when $d \leq .20$, medium when $.20 < d \leq .50$, large when $.50 < d \leq 1.0$, and very large when $d > 1.0$ [53]. Internal consistency was examined through Cronbach's alphas. To explore the normality of the scales' distributions, the following criteria were used: skewness, kurtosis, Kolmogorov-Smirnov Goodness-of-Fit Test ($N > 30$), steam and leaf diagrams and Q-Q plots. Given that most of the LPFS-SR scales did not follow a normal distribution in the community sample, the Kruskal-Wallis test with multiple pairwise comparisons was used to compare the three groups (PD sample, OD sample and community sample). In addition, the epsilon-squared ($\varepsilon^2$) estimate of effect size was obtained, in which $\varepsilon^2 < .01$ is negligible, $.01 \leq \varepsilon^2 < .04$ is weak, $.04 \leq \varepsilon^2 < .16$ is moderate, $.16 \leq \varepsilon^2 < .36$ is relatively strong, $.36 \leq \varepsilon^2 < .64$ is strong and $.64 \leq \varepsilon^2 < 1.00$ is very strong [53]. The capacity of the LPFS-SR total score to discriminate between the PD and the community samples; and between the PD and OD samples was examined through the ROC curve analysis. The factor structure of the Portuguese version of the LPFS-SR in the community and clinical samples was examined through the FACTOR software, in which the Hull method with unweighted least squares extraction and Promax rotation was applied, and the Hull scree-test value, the Comparative Fit Index (CFI) and the Non-normed Fit Index (NNFI or TLI) were used to test the goodness of fit. CFI and TLI above .95 indicate a good to excellent fit of the model. Additionally, to test closeness to unidimensionality, the Unidimensional Congruence (UniCo), the Explained Common Variance (ECV) and the Mean of Item Residual Absolute Loading (MIREAL) indexes were analysed. A UniCo score above .95, a ECV score above .85 and a MIREAL score below .30 suggest that the data can be treated as essentially unidimensional [51].

To analyse the similarity between factor results, Tucker's congruence coefficient [54] was used. A value in the range [.85, .94] indicates that the two compared factors are fairly similar, while a value equal to or higher than 0.95 signifies good similarity [49].

## Results

Table 1 presents the Cronbach's alphas ($\alpha$), means ($M$), standard deviations ($SD$), and Cohen's d between the USA sample [26] and the Portuguese community and clinical samples for the LPFS-SR domains and total.

In the community sample, descriptive statistics for the domains and total of the Portuguese version of the LPFS-SR were compared with Morey's original data [26] through Cohen's d. There were low ($\leq 0.2$) to medium effect sizes (0.2–0.5). These slight variations suggest that the Portuguese version of the LPFS-SR produces scores in the same range as those found with the original version of the LPFS-SR. Therefore, this study tends to contribute to the cross-cultural validation.

**Table 1. Cronbach's alphas (α), means (*M*), standard deviations (*SD*), and Cohen's *d* between the USA sample [26] and the Portuguese community and clinical samples for the LPFS-SR domains and total.**

|  | Morey (2017) (*N* = 293*) | | | Community (*N* = 282) | | | | Clinical (*N* = 188) | | |
|---|---|---|---|---|---|---|---|---|---|---|
|  | *M* | *SD* | α | *M* | *SD* | α | *d* | *M* | *SD* | α |
| Identity | 75.83 | 25.17 | .89 | 69.37 | 18.72 | .87 | .29 | 94.68 | 25.19 | .86 |
| Self-Direction | 53.37 | 20.19 | .88 | 50.24 | 13.46 | .82 | .18 | 68.92 | 20.65 | .86 |
| Empathy | 39.09 | 14.26 | .82 | 40.37 | 10.32 | .67 | -.10 | 50.50 | 13.91 | .65 |
| Intimacy | 64.09 | 22.99 | .88 | 55.59 | 15.02 | .80 | .44 | 74.20 | 21.44 | .82 |
| Total | 232.38 | 76.45 | .96 | 214.58 | 50.14 | .96 | .27 | 287.24 | 73.66 | .95 |

* USA community sample

*Note*. Small effect $d \leq .20$, medium effect size $.20 < d \leq .50$, large $.50 < d \leq 1.0$, and very large $d > 1.0$

Cronbach's alphas for the subcomponents of the LPFS-SR in the community and clinical samples were generally in the .80 range, with lower internal consistency noted for the empathy subscale. Notably, the alphas for the total score were .96 in the community sample (.96 in the USA community sample; [26]) and .95 in the clinical sample. Item-total correlations showed that most of the items correlated at acceptable levels ($> .30$) with their respective scale and the total score. It was noted that items 40, 44, 51, 61 and 76 in the community sample, and items 11, 14, 40, 44, 51, 61 and 76 in the clinical sample obtained negative item-total correlations. These are all reverse-keyed items that represent positive personality features, and are thus weighted negatively in computing the overall LPFS-SR total score.

Table 2 presents the LPFS-SR scales' means (*M*) and standard deviations (*SD*) in the Portuguese community sample, in the PD sample and in the OD sample.

The LPFS-SR scale means were higher in the PD sample, followed by the OD sample and finally by the community sample (Tables 1 and 2).

Tables 3–6 display the LPFS-SR scales' intercorrelations in the Portuguese community sample, in the clinical sample and sub-samples, the PD sample and in the OD sample, respectively.

The four LPFS-SR scales revealed strong intercorrelations ranging from .56 to .74 in the community sample, from .64 to .76 in the total clinical sample, from .55 to .72 in the PD sample, and from .66 to 76 in the OD sample. Scale to total correlations ranged from .80 to .90 in the community sample, from .82 to .91 in the total clinical sample, from .82 to .89 in the PD sample, and from .83 to .90 in the OD sample.

The Kruskal-Wallis H-test with multiple pairwise comparisons was used to compare the three groups (PD group, OD group and community sample).

Table 7 presents the Kruskal-Wallis test results in the three samples.

**Table 2. LPFS-SR scales' means (*M*) and standard deviations (*SD*) in the Portuguese personality disorder sample and in the other diagnoses sample.**

|  | Personality Disorder (*n* = 40) | | Other diagnoses (*n* = 148) | |
|---|---|---|---|---|
|  | *M* | *SD* | *M* | *SD* |
| Identity | 101.97 | 24.28 | 92.72 | 25.16 |
| Self-Direction | 74.92 | 21.59 | 67.23 | 20.14 |
| Empathy | 53.15 | 12.06 | 49.79 | 14.33 |
| Intimacy | 80.67 | 20.34 | 72.50 | 21.47 |
| Total | 309.58 | 71.37 | 281.47 | 73.42 |

**Table 3. LPFS-SR scales' intercorrelations in the Portuguese community sample.**

| | Community (N = 282) | | | | |
|---|---|---|---|---|---|
| | Identity | Self-Direction | Empathy | Intimacy | Total |
| Identity | 1.00 | | | | |
| Self-Direction | .74** | 1.00 | | | |
| Empathy | .60** | .65** | 1.00 | | |
| Intimacy | .62** | .56** | .62** | 1.00 | |
| Total | .90** | .85** | .80** | .82** | 1.00 |

** p < .01

**Table 4. LPFS-SR scales' intercorrelations in the total clinical sample.**

| | Clinical (N = 188) | | | | |
|---|---|---|---|---|---|
| | Identity | Self-Direction | Empathy | Intimacy | Total |
| Identity | 1.00 | | | | |
| Self-Direction | .76** | 1.00 | | | |
| Empathy | .69** | .64** | 1.00 | | |
| Intimacy | .72** | .71** | .70** | 1.00 | |
| Total | .91** | .88** | .82** | .89** | 1.00 |

** p < .01

**Table 5. LPFS-SR scales' intercorrelations in the Portuguese personality disorder sample.**

| | Personality Disorder (n = 40) | | | | |
|---|---|---|---|---|---|
| | Identity | Self-Direction | Empathy | Intimacy | Total |
| Identity | 1.00 | | | | |
| Self-Direction | .70** | 1.00 | | | |
| Empathy | .63** | .55** | 1.00 | | |
| Intimacy | .72** | .60** | .56** | 1.00 | |
| Total | .88** | .82** | .82** | .89** | 1.00 |

** p < .01

**Table 6. LPFS-SR scales' intercorrelations in the Portuguese other diagnoses sample.**

| | Other diagnoses (n = 148) | | | | |
|---|---|---|---|---|---|
| | Identity | Self-Direction | Empathy | Intimacy | Total |
| Identity | 1.00 | | | | |
| Self-Direction | .76** | 1.00 | | | |
| Empathy | .70** | .66** | 1.00 | | |
| Intimacy | .71** | .71** | .72** | 1.00 | |
| Total | .90** | .89** | .83** | .88** | 1.00 |

** p < .01

**Table 7. Mean ranks, Independent Samples Kruskal-Wallis ($\chi^2$) and size effects ($\varepsilon^2$) in the community, personality disorder and other diagnoses samples.**

| LPFS-SR scales | Samples | Mean ranks | $\chi^2$ | $p$ | $\varepsilon^2$ |
|---|---|---|---|---|---|
| Identity | Community | 174.37 | 115.26 | < .001 | .26 |
| | Personality Disorders | 341.09 | | | |
| | Other Diagnoses | 298.53 | | | |
| Self-Direction | Community | 177.05 | 104.99 | < .001 | .23 |
| | Personality Disorders | 336.83 | | | |
| | Other Diagnoses | 295.00 | | | |
| Empathy | Community | 186.22 | 65.75 | < .001 | .15 |
| | Personality Disorders | 317.96 | | | |
| | Other Diagnoses | 276.72 | | | |
| Intimacy | Community | 175.56 | 88.71 | < .001 | .20 |
| | Personality Disorders | 327.67 | | | |
| | Other Diagnoses | 280.23 | | | |
| Total | Community | 156.76 | 103.76 | < .001 | .26 |
| | Personality Disorders | 308.16 | | | |
| | Other Diagnoses | 268.71 | | | |

*Note.* $\varepsilon^2$; negligible effect: < .01, weak effect: .01 - .04, moderate effect: .04 - .16, relatively strong effect .16 - .36, strong effect .36 - .64, very strong effect: .64–1.00

The PD sample, the OD sample and the community sample differed in all the LPFS-SR scales and total scores ($p < .001$), with relatively strong effects in all the dimensions, except in the empathy scale in which the epsilon-squared estimate of effect size was moderate.

Multiple pairwise comparisons also revealed that in all the LPFS-SR scales and total scores, the community sample differed significantly from the OD group ($p < .001$) and the PD group ($p = .001$). The clinical samples did not differ in the LPFS-SR scales, although the PD mean ranks were higher than the OD mean ranks.

ROC analyses were used to examine the capacity of the LPFS-SR total score to discriminate between the samples. Considering that the LPFS-SR is intended to specifically assess the core personality features of PD, only the LPFS-SR total score ROC discriminative capacity between the PD and community samples and between the PD and OD samples were reported. The optimal cut-off points were selected such that the sum of sensitivity and specificity was maximized, thus the optimal score would produce few incorrect decisions under conditions of equal a priori probabilities. Classification results using the optimal cut-off score obtained by the ROC analysis are provided.

The LPFS-SR total score ROC discriminative capacity between the PD and community samples demonstrated a significant area-under-the-curve of .86 ($p < .001$; 95% CI: .78-.94), indicating that the LPFS-SR total score significantly discriminated between the community and the PD samples (see Fig 1). The optimal cut-off was a total score of 272.00, corresponding to a sensitivity (true positive rate) of 75% and a specificity (true negative rate) of 89% (1-.11). This cut-off accurately classified 88% of the cases, 89% of the community sample and 75% of the PD sample.

Fig 2 shows that the LPFS-SR total score demonstrated a significant but small degree of discrimination between the PD and OD samples. The area-under-the-curve was of .63 ($p = .027$; 95% CI: .52-.74). The optimal cut-off was a total score of 294.50, corresponding to a sensitivity (true positive rate) of 66% and a specificity (true negative rate) of 59% (1-.41). This cut-off accurately classified 60% of the cases, 59% of the OD sample and 66% of the PD sample.

In line with Hemmati et al. [50], the Factor software [51], was used to address the unidimensionality of the LPFS-SR in the community and in the clinical samples. In the community sample, preliminary data checks revealed that the Kaiser-Meyern (KMO) measure of sampling

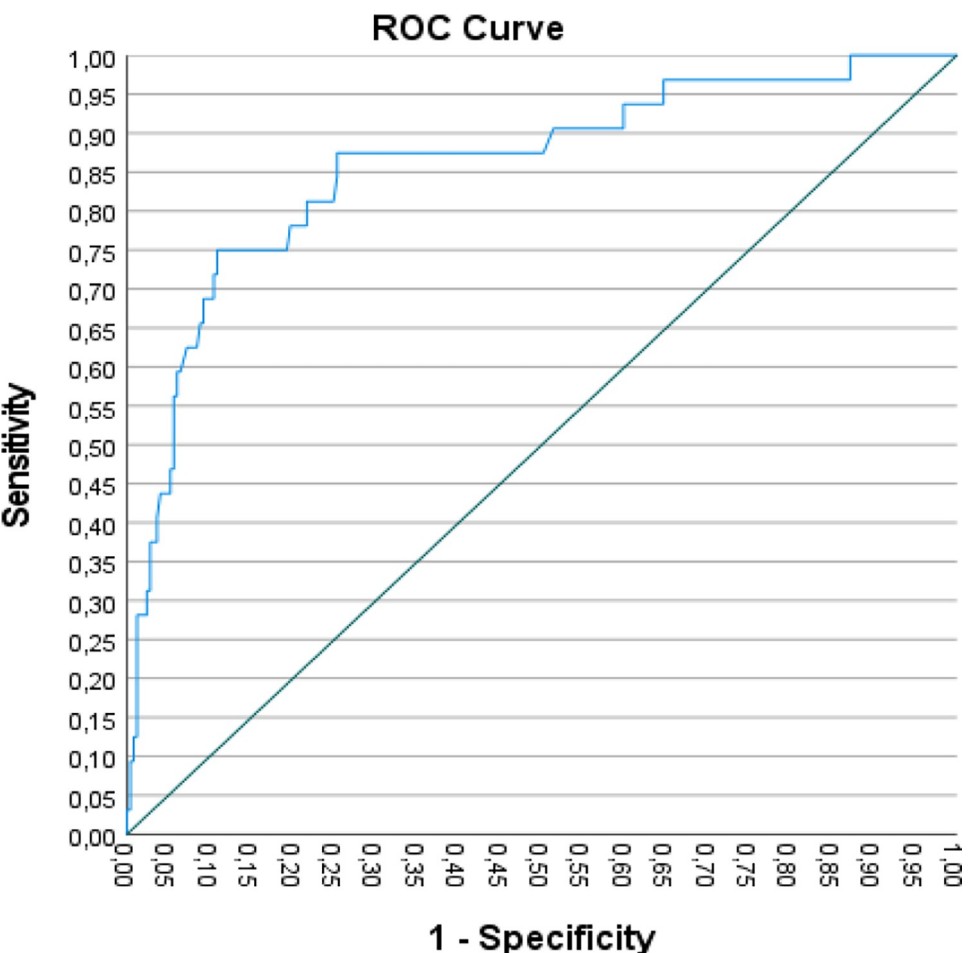

**Fig 1. Receiver Operator Characteristic (ROC) curve for the LPFS-SR in differentiating the community and PD samples.**

adequacy was good (.87). The results are consistent with those obtained by Hemmati et al. [50] and support a one-dimensional factor solution: the Hull method with unweighted least squares, scree-test value = 21.69, CFI = .999, NNFI (TLI) = 1.038, UniCo = .959, ECV = .871 and MIREAL = .179. Also, the strong intercorrelations among the four LPFS-SR scales and the LPFS-SR total score support the unidimensional nature of the measure (see Table 3). Moreover, the fact that the eigenvalue of the first factor (17.05; 21%) is much larger than the eigenvalue of the second factor (4.91; 6%) further suggests that the 80 items of the LPFS-SR can be treated as essentially unidimensional.

In the clinical sample, preliminary data checks also showed that the Kaiser-Meyern (KMO) measure of sampling adequacy was good (.81). The results are in line with those obtained for the community sample, although the reduced number of psychiatric patients in the clinical sample led to more modest index values, but supported a one-dimensional factor solution: the Hull method with unweighted least squares and Promax rotation, scree-test value = 20.15, CFI = .982, NNFI (TLI) = .987, UniCo = .952, ECV = .875 and MIREAL = .213. Also, the strong intercorrelations among the four LPFS-SR scales and the LPFS-SR total score support the uni-dimensional nature of the measure (see Table 4). Moreover, the fact that the eigenvalue of the

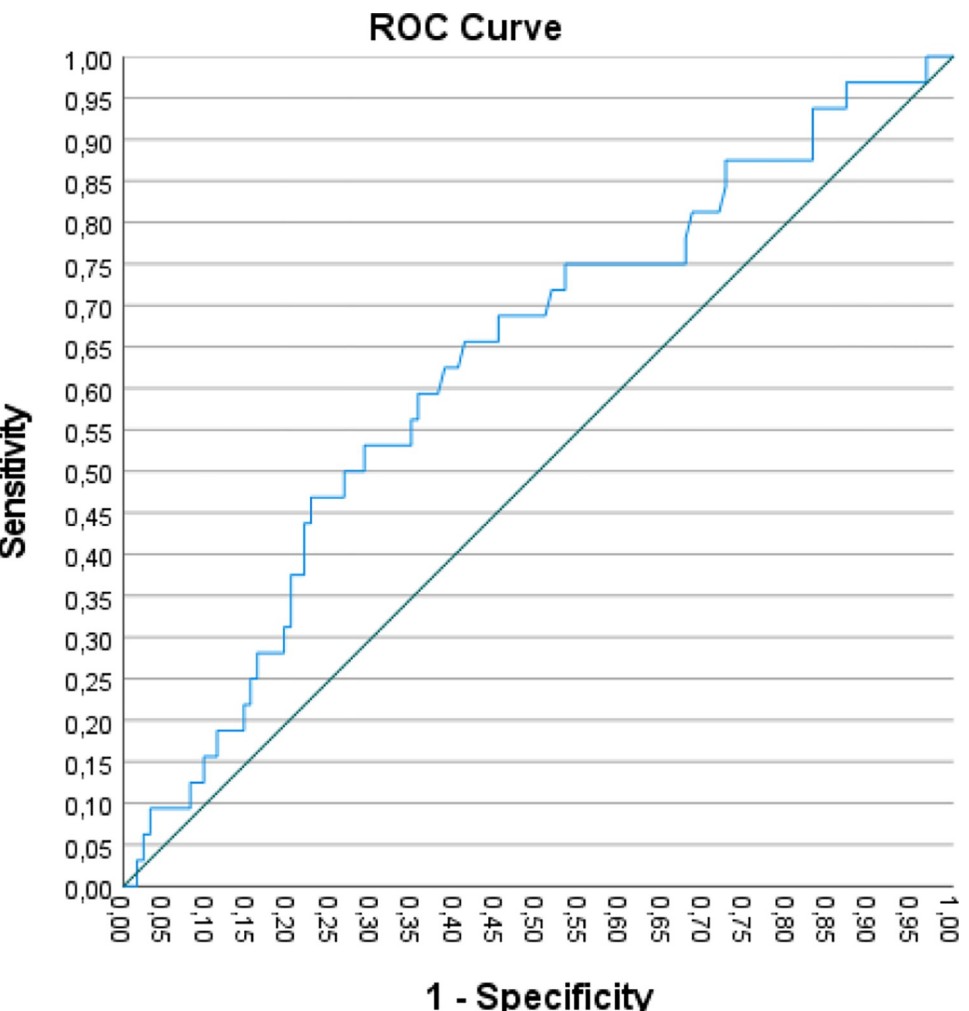

Fig 2. Receiver Operator Characteristic (ROC) curve for the LPFS-SR in differentiating the PD and OD samples.

first factor (18.66; 23%) is much larger than the eigenvalue of the second factor (5.49; 7%) further suggests that the 80 items of the LPFS-SR can be treated as essentially unidimensional.

The similarity between the extracted factors in the community and clinical samples, and international studies with available factors results, was analysed using Tucker's congruence coefficient (φ). We compared the factor results of the Portuguese community sample, the Portuguese clinical sample, and the item factor loadings of Bliton et al. [55].

φ(Community vs. Clinical samples) = .98, φ(Community sample vs. Bliton study) = .94 and φ(Clinical sample vs. Bliton study) = .94. Thus, the factor structures for the community and clinical samples exhibit good similarity; community factor loadings and Bliton study factor loadings are fairly similar; clinical sample and Bliton study factor results are also fairly similar.

## Discussion

The aim of this study was to contribute to the cross-cultural validation of the LPFS-SR in Portuguese community and clinical samples and to address the LPFS-SR's potential for

distinguishing between non-clinical participants and clinical participants with respect to the severity of personality dysfunction, as well as the dimensionality of the LPFS-SR construct.

The Portuguese version of the LPFS-SR showed adequate psychometric properties. The results for internal consistency were in line with those obtained with the original test [26, 55], with the Persian version of the test [50], in which both a clinical and community sample were studied, and with its Polish version, in a community sample [47]. In the Polish adaptation, the alpha of the empathy scale was akin to the Portuguese alpha in both samples, that is lower than the other scales' alphas.

As for its validity, the results showed that the Portuguese LPFS-SR was able to differentiate the clinical and community samples. As expected for a measure of personality pathology severity, the community sample yielded significantly lower results in all the LPFS-SR scales than both clinical samples. Moreover, considering that the LPFS-SR is intended to assess personality dysfunction severity in the interest of diagnosing PD, the Portuguese version of the LPFS-SR should be able to screen and predict a PD diagnosis. The ROC analysis showed that the highest LPFS-SR total score discriminative capacity was obtained between the PD and community samples (AUC = .86; $p < .001$; optimal cut-off of 272.00, corresponding to a sensitivity of 75% and a specificity of 89%), however, although our PD sample was small and had comorbidities, the LPFS-SR total score discriminative capacity between the PD and the OD sample was also significant (AUC = .63; $p = .027$). It should be noted that the LPFS model not only seeks to describe core features of PD, but may also characterize impairments in self and interpersonal functioning underlying many forms of psychopathology [45, 56]. Some have proposed that the LPFS construct is closely aligned with a general factor of psychiatric severity, such as the p factor [57], reflecting the overall level of impairment or dysfunction [34, 45]. Given that all psychopathology is mediated by the individual's subjective self-other experience, this may account for its less discriminative capacity between PDs and non-PD pathology.

In keeping with Hemmati et al. [50] and Łakuta et al.'s [47] findings, in this study the Cronbach's alpha values for the community and clinical samples were high, above or equal to .80 and .82, respectively, except for the empathy scale. Moreover, all the LPFS-SR scales showed strong intercorrelations in both samples. Finally, the EFA and CFA indexes obtained in this study supported a unidimensional structure for the LPFS-SR in the community and clinical sample [26, 34, 46, 55]. Therefore, the current study supports the premise that the LPFS-SR construct lays in a continuum that captures the degree of severity of personality dysfunction.

Despite the promising and convergent results that contribute to the validation of LPFS-SR usage in clinical contexts, the validation of the Portuguese version of the LPFS-SR is still in progress. Conducting validity studies (for convergent validity data, please refer to [36]) and performing scale reliability analyses with a test-retest design, are essential future endeavours to ensure the accuracy of the LPFS-SR for use in clinical practice and scientific research. Other limitations are related to the small size of the samples and the comorbid diagnoses which, although reflecting the reality of the psychiatric population, may have accounted for the less clearly interpretable differences between the PD and the OD samples. Although each diagnosis assignment relied on the clinical evaluation of several psychiatrists per mental health institution, the fact that no structured clinical interview was used may have undermined diagnostic reliability and contributed to less expressive differences between the PD and OD samples. On the other hand, the fact that in the current study, besides the clinical, a community sample was used instead of solely a college sample, reinforces the cross-cultural validation of the LPFS-SR. Moreover, on the 10th anniversary of the publication of the AMPD [1], and in line with Zimmermann's appeal [41], this study offers a modest contribution to a constructive discussion regarding the strengths and limitations of Criterion A, namely supporting its one-dimensionality [20, 26, 46].

As future directions and considering the cross-validation of the LPFS-SR, it would be interesting to compare Portuguese results with those obtained in other Portuguese speaking countries. The study of the discriminative capacity of the LPFS-SR with enlarged PD samples and other clinical and non-clinical samples is one of the most anticipated future developments for this study as it may be another step in the validation of the DSM-5 AMPD model, eventually contributing to its definitive inclusion in the official DSM-5 PD classification.

## Supporting information

**S1 File.**
(PDF)

**S2 File.**
(PDF)

**S3 File.**
(XLSX)

## Acknowledgments

To Tania Gregg for specialized assistance in the proofreading of this paper. To all the study participants without whom this study would not have been possible and to whom it is dedicated.

## Author Contributions

**Conceptualization:** Rute Pires.

**Data curation:** Rute Pires, Joana Henriques-Calado, Bruno Gonçalves.

**Formal analysis:** Rute Pires, Joana Henriques-Calado, Ana Sousa Ferreira, Bruno Gonçalves.

**Funding acquisition:** Rute Pires, Joana Henriques-Calado, Bruno Gonçalves.

**Investigation:** Rute Pires, Joana Henriques-Calado, Bruno Gonçalves.

**Methodology:** Rute Pires, Joana Henriques-Calado, Ana Sousa Ferreira, Bruno Gonçalves.

**Project administration:** Rute Pires, Joana Henriques-Calado, Bruno Gonçalves.

**Resources:** Rute Pires, Joana Henriques-Calado, João Gama Marques, Ana Ribeiro Moreira, Bernardo C. Barata, Marco Paulino, Bruno Gonçalves.

**Software:** Rute Pires, Ana Sousa Ferreira.

**Supervision:** Rute Pires, Joana Henriques-Calado, Ana Sousa Ferreira, Leslie Morey, Bruno Gonçalves.

**Validation:** Rute Pires, Joana Henriques-Calado, Ana Sousa Ferreira, Leslie Morey, Bruno Gonçalves.

**Visualization:** Rute Pires.

**Writing – original draft:** Rute Pires.

**Writing – review & editing:** Rute Pires, Joana Henriques-Calado, Ana Sousa Ferreira, João Gama Marques, Ana Ribeiro Moreira, Bernardo C. Barata, Marco Paulino, Leslie Morey, Bruno Gonçalves.

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
