## [Decision Letter · Decision Letter 0]

7 Sep 2023

PONE-D-23-22309The Portuguese version of the self-report form of the DSM-5 Level of Personality Functioning Scale (LPFS-SR) in a community and clinical samplePLOS ONE

Dear Dr. Pires,

Thank you for submitting your manuscript to PLOS ONE. After careful consideration, we feel that it has merit but does not fully meet PLOS ONE’s publication criteria as it currently stands. Therefore, we invite you to submit a revised version of the manuscript that addresses the points raised during the review process. This is a very interesting and good quality article, There are several aspects to improve as pointed by reviewers. Please, consider to publish the Portuguese translated version as an anex to the article, instead of the approval by scientific council of your institution.

We look forward to receiving your revised manuscript.

Kind regards,

Paulo Alexandre Azevedo Pereira Santos, PhD

Academic Editor

PLOS ONE

“This work received national funding from the FCT – Fundação para a Ciência e a Tecnologia, I.P [Foundation for Science and Technology], through the Research Center for Psychological Science of the Faculty of Psychology, University of Lisbon (UIDB/04527/2020; UIDP/04527/2020) and the Business Research Unit, BRU-IUL (UIDB/00315/2020).”

Additional Editor Comment:

This is a very interesting and good quality article, There are several aspects to improve as pointed by reviewers.

Please, consider to publish the Portuguese translated version as an anex to the article, instead of the approval by scientific council of your institution.

Reviewers' comments:

Reviewer's Responses to Questions

**Comments to the Author**

1. Is the manuscript technically sound, and do the data support the conclusions?

Reviewer #1: Yes

Reviewer #2: No

2. Has the statistical analysis been performed appropriately and rigorously? 

Reviewer #1: Yes

Reviewer #2: I Don't Know

3. Have the authors made all data underlying the findings in their manuscript fully available?

Reviewer #1: No

Reviewer #2: No

4. Is the manuscript presented in an intelligible fashion and written in standard English?

Reviewer #1: Yes

Reviewer #2: No

5. Review Comments to the Author

Reviewer #1: This manuscript reports on the psychometric properties of a Portuguese translation of the Levels of Personality Functioning Scales (LPFS-SR), self-report version developed by Morey (2017). The study provides useful information for users of the LPFS in Portuguese-speaking populations, and it provides a needed cultural analysis of this relatively new clinical instrument. I have some suggestions for the authors to consider in further revisions of the manuscript.

1. The number of ROC analyses reported in the paper seems excessive. I see the merit in examining basic contrasts of the means for the LPFS elements and total score between the community and clinical samples. The LPFS is intended to assess core personality functions in the interest of diagnosing personality disorder (PD); thus, the ROC analysis contrasting clinical patients with PD versus clinical patients without PD is considerably more relevant to the study aims than the other contrasts. What is the relevance of a cut score for discriminating community adults from patients who do not have PD on a measure specifically designed for the assessment of PD? Moreover, nearly all clinical measures will discriminate community adults from clinical patients, so the comparisons of different diagnostic groups are most useful for validating the LPFS.

2. Comparing the community adults from Portugal to the US sample presented in Morey (2017) would effectively address the stated aim to evaluate the “cross-cultural validity.”

3. Regarding the community sample described in Lines 120-122. What incentives, if any, were offered to these acquaintances of university students? Why did they participate? How did they access the study protocol? Were the instruments administered remotely, online, or in person?

4. Likewise, I think some further details on the mode of data collection for the clinical patients would be helpful. Was the LPFS-SR administered only for research purposes? Or was it part of a clinical assessment that informed care of these patients?

5. The decision to investigate dimensionality only with the community sample, defended in Lines 302-304, is questionable. I do not follow these arguments, and given the intended use of the LPFS-SR in clinical settings, the inclusion of patient respondents seems essential.

6. As for future directions, it would be interesting to administer this translation in Brazil to see if the psychometric properties are comparable to those from persons in Lisbon.

7. Line 64: It is an overstatement to say “few studies” have focused on Criterion A. I think the authors mean to say that fewer studies have addressed Criterion A relative to Criterion B.

8. Line 87: I disagree that adequate psychometric properties “guarantee” a deeper understanding of something as complex as personality pathology. They are a necessary start in that direction.

9. Lines 68-70: I don’t understand this sentence about Quilty’s conclusion. The point seems circular.

10. Lines 95-96: I assume “the original data” refer to information about the derivation of the LPFS-SR in Morey (2017). If so, that should be stated explicitly here.

11. Line 108: I think the word “call” is meant here, not “claim.”

12. Line 363: I think the word “affirms” or “supports” is more apt here, not “sustains.”

Reviewer #2: This is an interesting paper on the psychometric properties of the LPFS-SR, which is among the best self-report questionnaires for the assessment of the A criterion of the Alternative DSM-5 Model for Personality Disorders, maybe the best. I have no comments on the design and implementation of the study. However, I found the reasoning in the introduction and discussion quite flawed from time to time.

Line 63-67: I am not sure if this statement is correct. First, by the time of publication of DSM-5 a decade ago, the LPFS was a brand-new scale whereas the PID-5 already existed in 2013 and had been evaluated extensively. This is probably the main reason why there are fewer studies on the LPFS than on the PID-5. Moreover, the decade after the publication of DSM-5 witnessed a bourgeoning of research on the LPFS so it is certainly not correct to state that only a few studies have focused on the LPFS, e.g., (Zimmermann et al., 2019). Moreover, the statement on line 66 authors suggests that the LPFS should be assessed by clinicians whereas this is not required for the trait model. I really could not find this information in the DSM-5. The trait model also plays a role in the diagnostic process, requiring the use of a structured diagnostic interview.

Line 68-70. The authors make it clear that personality impairment should be assessed using a clinician-rated instrument. However, the current study uses a self-report instrument. How could this be reconciled? The concluding sentence of the first paragraph highlights the importance of clinician-rated instruments, giving the impression that the current study is on a clinical interview. Since this is not the case, the focus of the first paragraph should be redirected.

Line 81. The authors refer to only one study evaluating the LPFS-BF and this study concerns the first version of the LPFS-BF. The second version has better psychometric properties as supported by several studies. Please correct.

Line 79-92. I think it could be emphasized more that the LPFS-SR is the only self-report instrument that takes both the vertical and horizontal structure of the LPFS into account.

Line 93-111. I had troubles following along with the aims of the study because this section is merely a combination of research aims, methodological issues such as sample descriptions, the rationale for the study, and empirical findings. I hope the authors will be able to formulate the research questions more clearly. Also, what are “the original data”?

Line 90-92. Do the authors suggest that maladaptive personality functioning is distinct from maladaptive personality content?

Line 106-111. The authors seem to suggest that there is no doubt that the LPFS-SR reflects an unidimensional factor whereas Sleep and colleagues are totally wrong. These formulations poorly reflect the discussion between Morey and Sleep about the dimensionality of the LPFS-SR. I think it would be wise to explain this discussion in more detail since it provides the main rationale for the current study. I also recommend to move this discussion to the introduction.

Line 148-164: I could not find any information about personality disorder assessment. How were PD diagnoses and symptom diagnoses assessed? If no structured clinical interviews were used, this should be mentioned as a limitation in the discussion. It is well known that clinical assessment of PDs without making use of structured diagnostic interviews, has poor diagnostic reliability. Thus, the lack of differences between the PD sample and non-PD sample could as well be due to poor assessment procedures.

Overall, I think the manuscript could benefit from some more text-editing to make it more reader-friendly at some places. For instance, regarding the explanation of the translation process on line 156-164, the authors could write “obtained” instead of “requested” and “translator” instead of “author of the translation”. Moreover, it is not clear whether the back-translation was done by one person (a native English speaker who also was a professional translator) or two persons (a native English speaker and a professional translator).

Line 182-188: I am not acquainted with the Factor software and have never seen other papers using this program. This might be due to lack of expertise on my part but it could also be due to the infrequent use of this program in the PD field. Therefore, I think it would be appropriate to give a more extensive explanation of this program in the Methods chapter. I would also recommend to restrict the explanation of the statistical procedures to the “Data analysis” section, i.e., move the explanation that “CFA and EFA are congregated” to the Methods chapter. It should be noted that, since I am not a psychometrician and do not have any background knowledge about the Factor package, my ability to assess the adequacy of the statistical analyses is limited.

Line 321-343: In scientific papers, it is common to provide several explanations for unexpected findings. I have already mentioned the possibility of poor diagnostic reliability. Could there be other explanations for the finding that the LPFS-SR could not make a distinction between the two clinical samples? Such a discussion is more interesting and informative than just a repetition of the results and a conclusion that more research is needed.

Line 340-343: the information in the section is not correct. Self pathology and interpersonal problems are the core features of PD, by definition.

Line 344-355. This is an important section, which needs thorough revision. First, I couldn’t find back the proposed threshold values in the paper of Morey et al. (Morey, 2017). I neither could find any information about the interpretative criterion in this paper. Could the authors give some more explanation of these “threshold values” and “interpretative criterion”? I also had difficulties understanding how cultural factors could have increased the threshold in the Iranian sample and why we should address interpretability of the LPFS-SR scores. Could this be explained more clearly? To me, it seems that sample characteristics are more important than possible cultural factors. After all, the sample of Hemmati et al. consisted of inpatients, most of whom were diagnosed with either borderline PD or antisocial PD. Moreover, I did not manage to understand the last sentence of this paragraph. Could the authors’ point be explained more clearly?

Line 359: It is a common error to interpret a large Chronbach’s alpha value as proof for unidimensionality. The authors could easily avoid this error by focusing more on the results of the factor analyses.

Line 365-381: The section about limitations should have a stronger on the very limitations rather than repeating the results. How could these imitations have influenced the results? For instance, what kind of errors could be associated with the small size of the PD sample? And how could comorbidity have influenced the results? (Is this really a limitation? A clinical sample without any comorbidity would be a rarity and not very representative.) If it is correct that no structured diagnostic instruments were used, this should also be mentioned as a limitation. Finally, I am not sure if it is wise to write “Criterion A’s opponents”. In my view, this formulation has a rather negative connotation in this context. I assume that the authors are referring to Sleep & colleagues. Sure, these authors are critical to the view that the LPFS is a unidimensional construct but I really cannot see that they are opposing the A criterion.

Morey, L. C. (2017). Development and Initial Evaluation of a Self-Report Form of the DSM-5 Level of Personality Functioning Scale. Psychological Assessment, 27(10), 1302.

Zimmermann, J., Kerber, A., Rek, K., Hopwood, C. J., & Krueger, R. (2019). A brief but comprehensive review of research on the Alternative DSM-5 Model for Personality Disorders. Current Psychiatry Reports, 21(92). https://doi.org/10.1007/s11920-019-1079

6. PLOS authors have the option to publish the peer review history of their article (what does this mean?). If published, this will include your full peer review and any attached files.

Reviewer #1: **Yes: **John E Kurtz

Reviewer #2: **Yes: **Benjamin Hummelen

---

## [Author Response · Author response to Decision Letter 0]

11 Nov 2023

The Portuguese version of the self-report form of the DSM-5 Level of Personality Functioning Scale (LPFS-SR) in a community and clinical sample [PONE-D-23-22309] 

RESPONSE TO REVIEWERS

REVIEWER 1

We would like to thank Reviewer 1 for considering our manuscript entitled The Portuguese version of the self-report form of the DSM-5 Level of Personality Functioning Scale (LPFS-SR) in a community and clinical sample [Manuscript ID: PONE-D-23-22309] for revision. Reviewer 1's comments have provided detailed and constructive feedback on our manuscript which has given us an outer perspective of our work and helped us to think critically and, we believe, to improve the paper. We have now revised the manuscript according to Reviewer 1's suggestions and we hope that we have been able to address all the issues raised and comments. Please note that in the document “Revised Manuscript with Track Changes” the changes and insertions are highlighted (green colour). In the following lines we respond to the points addressed, our comments and inserted text are in green, and follow the order in which they were raised. 

1. The number of ROC analyses reported in the paper seems excessive. I see the merit in examining basic contrasts of the means for the LPFS elements and total score between the community and clinical samples. The LPFS is intended to assess core personality functions in the interest of diagnosing personality disorder (PD); thus, the ROC analysis contrasting clinical patients with PD versus clinical patients without PD is considerably more relevant to the study aims than the other contrasts. What is the relevance of a cut score for discriminating community adults from patients who do not have PD on a measure specifically designed for the assessment of PD? Moreover, nearly all clinical measures will discriminate community adults from clinical patients, so the comparisons of different diagnostic groups are most useful for validating the LPFS.

Thank you for this outer perspective of our work which, we believe, has helped us improve the paper. We have followed the suggestion and have reported only the ROC contrasts between the PD sample and the community sample and the PD sample and the other clinical sample.

We agree that these contrasts are the most relevant for the validity of a measure that pertains to assess features relevant to a PD diagnosis. 

Thus, we have decided to exclude from the text the reference to the optimal cut-off to discriminate between the clinical and the non-clinical population. The decision to mention it derived from Hemmati et al.'s study, however since this was also questioned by Reviewer #2, the text has been reformulated.

Please see lines: 227-229; 314-318; 401-415.

2. Comparing the community adults from Portugal to the US sample presented in Morey (2017) would effectively address the stated aim to evaluate the “cross-cultural validity.”

Thank you for this suggestion which, we believe, enriches the manuscript, and, indeed, contributes to LPFS-SR cross-cultural validity. To compare the community adults from Portugal to the US sample presented in Morey (2017) we used the Cohen’s d and obtained low to medium effect sizes that suggest slight variations between the Portuguese version and Morey’s original data (2017). Please see lines 215-218; 249-257.

3. Regarding the community sample described in Lines 120-122. What incentives, if any, were offered to these acquaintances of university students? Why did they participate? How did they access the study protocol? Were the instruments administered remotely, online, or in person?

Thank you for the opportunity to clarify the sampling procedures. Please see lines 150-158.

4. Likewise, I think some further details on the mode of data collection for the clinical patients would be helpful. Was the LPFS-SR administered only for research purposes? Or was it part of a clinical assessment that informed care of these patients?

Thank you for the opportunity to clarify the sampling procedures. Please see lines 159-160.

5. The decision to investigate dimensionality only with the community sample, defended in Lines 302-304, is questionable. I do not follow these arguments, and given the intended use of the LPFS-SR in clinical settings, the inclusion of patient respondents seems essential.

We have followed Reviewer #1's suggestion and have also explored dimensionality in the clinical sample. Please see lines 229-238; Table 3B; 362-384; 419-423.

6. As for future directions, it would be interesting to administer this translation in Brazil to see if the psychometric properties are comparable to those from persons in Lisbon.

Thank you for the suggestion which has been incorporated in the paper as a future direction. Please see lines 438-440.

7. Line 64: It is an overstatement to say “few studies” have focused on Criterion A. I think the authors mean to say that fewer studies have addressed Criterion A relative to Criterion B.

Thank you for the correction. Please see lines 60-62.

8. Line 87: I disagree that adequate psychometric properties “guarantee” a deeper understanding of something as complex as personality pathology. They are a necessary start in that direction.

Thank you for the correction. Please see line 89.

9. Lines 68-70: I don’t understand this sentence about Quilty’s conclusion. The point seems circular.

We agree that the sentence is not clear. Given that our manuscript does not concern clinician rated measures, there is no point in mentioning that they should be improved. We have deleted the sentence.

10. Lines 95-96: I assume “the original data” refer to information about the derivation of the LPFS-SR in Morey (2017). If so, that should be stated explicitly here.

Thank you. Please see line 130.

11. Line 108: I think the word “call” is meant here, not “claim.”

Thank you. Please see line 106.

12. Line 363: I think the word “affirms” or “supports” is more apt here, not “sustains.”

Thank you. Please see line 421.

REVIEWER 2

We would like to thank Reviewer 2 for considering our manuscript entitled The Portuguese version of the self-report form of the DSM-5 Level of Personality Functioning Scale (LPFS-SR) in a community and clinical sample [Manuscript ID: PONE-D-23-22309] for revision. Reviewer 2's comments have provided detailed feedback on our manuscript which, we hope, has given us an opportunity to improve it. Please note that in the document “Revised Manuscript with Track Changes” the changes and insertions are highlighted (green colour). In the following lines we respond to the points addressed, our comments and inserted text are in green, and follow the order in which they were raised. 

Line 63-67: I am not sure if this statement is correct. First, by the time of publication of DSM-5 a decade ago, the LPFS was a brand-new scale whereas the PID-5 already existed in 2013 and had been evaluated extensively. This is probably the main reason why there are fewer studies on the LPFS than on the PID-5. Moreover, the decade after the publication of DSM-5 witnessed a bourgeoning of research on the LPFS so it is certainly not correct to state that only a few studies have focused on the LPFS, e.g., (Zimmermann et al., 2019). 

Thank you for the opportunity to clarify the sentence. It was certainly not our intention to state that there were only a few studies with the LPFS. However, over the last decade, and compared to the studies focusing on the PID-5, there have been fewer studies with the LPFS than with the PID-5.

As reported by Zimmerman et al. (2019), 85% of the published studies on the AMPD addressed Criterion B while only 15% addressed Criterion A, and this may be explained by the original lack of a self-report measure for Criterion A (Le Corff et al., 2022).

We hope that the following reformulation clarifies the sentence and what we were trying to highlight:

Lines 59-62: “Notwithstanding the centrality of personality dysfunction in the diagnosis of PD, fewer studies have focused on the level of personality functioning (Criterion A) relative to traits (Criterion B) over the last decade.” 

- Zimmerman, J., Kerber, A., Rek, K., Hopwood, C. J., & Krueger, R. F. (2019). A brief but comprehensive review of research on the Alternative Model for Personality Disorders. Current Psychiatric Reports, 21(9), 92. https://doi.org/10.1007/s11920-019-1079-z

- Le Corff, Y., Aluja, A., Rossi, G., Lapalme, M., Forget, K., García, L. F., & Rolland, J. P. (2022). Construct Validity of the Dutch, English, French, and Spanish LPFS-BF 2.0: Measurement Invariance Across Language and Gender and Criterion Validity. Journal of Personality Disorders, 36(6), 662-679. https://doi.org/10.1521/pedi.2022.36.6.662

Moreover, the statement on line 66 authors suggests that the LPFS should be assessed by clinicians whereas this is not required for the trait model. I really could not find this information in the DSM-5. The trait model also plays a role in the diagnostic process, requiring the use of a structured diagnostic interview.

In line 66, we were trying to say that, compared to clinician rating scales, self-report measures, like the LPFS-SR, facilitate research.

As clinicians we agree that the diagnostic process should ultimately rely on clinical expertise. However, psychometric instruments can be useful in the process of collecting information about a patient. In the DSM-5 (APA, 2013), p. 774, there is a section regarding assessment of pathological traits where it is stated that this assessment “is facilitated by the use of formal psychometric instruments designed to measure specific facets and domains of personality. For example, the personality trait model is operationalized in the Personality Inventory for DSM-5 (PID-5), which can be completed in its self-report form by patients and in its informant-report form by those who know the patient well (e.g., a spouse).” 

Please inform us if any other change is required so that the sentence is properly understood:

“Notwithstanding the centrality of personality dysfunction in the diagnosis of PD, fewer studies have focused on the level of personality functioning (Criterion A) relative to traits (Criterion B) over the last decade. This is most likely due to the fact that the DSM-5 has proposed a clinician-rated scale for its characterization (LPFS; [19]), which is less suitable for research purposes than self-report measures” (Lines 59-64).

Line 68-70. The authors make it clear that personality impairment should be assessed using a clinician-rated instrument. However, the current study uses a self-report instrument. How could this be reconciled? The concluding sentence of the first paragraph highlights the importance of clinician-rated instruments, giving the impression that the current study is on a clinical interview. Since this is not the case, the focus of the first paragraph should be redirected.

Both Reviewers found the sentence [“Although little research has explored the psychometric properties of the AMPD clinician-rated measures, Quilty et al. [20] have recently concluded that as far as personality impairment is concerned, finer grained clinician-rated measures may be required.”] unclear. Given that it was not primarily related to our manuscript’s contents, it has been deleted. What we meant to say was that personality dysfunction may be assessed with clinician-rated measures or self-report measures. Our perspective is that self-reports facilitate research, therefore our manuscript is related to the LPFS-SR and not to a clinician-rated measure. However, personality dysfunction may also be assessed with clinician-rated measures, even if Quilty et al. (2021) consider that finer grained clinician-rated measures are also needed to achieve that goal.

Line 81. The authors refer to only one study evaluating the LPFS-BF and this study concerns the first version of the LPFS-BF. The second version has better psychometric properties as supported by several studies. Please correct.

Thank you for the opportunity to clarify the reason why we referred only to the first study concerning the LPFS-BF (Hutsebaut et al. 2016). We are aware that after the publication of the LPFS-SR (Morey, 2017), there was an update of the LPFS-BF which has seen several developments (Weekers et al., 2019; Weekers et al., 2022; Le Corff et al., 2022, among others). However, our study concerns the LPFS-SR whose development is a response to the fact that the existing measures at the time of the LPFS-SR development were not strictly aligned with the LPFS descriptors presented in the DSM-5. In the case of the first version of the LPFS-BF (Hutsebaut et al. 2016), the only one available at that time, Morey (2017) also refers to its limited internal consistency.

We have reformulated the sentence to highlight the fact that the LPFS-SR (Morey, 2017) was developed to respond to the need for a sufficiently detailed measure that would enable researchers to closely examine the particular strengths and weaknesses of the description of global personality pathology provided in the AMPD.

Please see lines 78-82.

- Le Corff, Y., Aluja, A., Rossi, G., Lapalme, M., Forget, K., García, L. F., & Rolland, J. P. (2022). Construct Validity of the Dutch, English, French, and Spanish LPFS-BF 2.0: Measurement Invariance Across Language and Gender and Criterion Validity. Journal of Personality Disorders, 36(6), 662-679. https://doi.org/10.1521/pedi.2022.36.6.662

- Weekers, L. C., Hutsebaut, J., & Kamphuis, J. H. (2019). The level of Personality Functioning Scale-Brief Form 2.0: Update of a brief instrument for assessing level of personality functioning. Personality and Mental Health, 13(1), 3–14. https://doi.org/10.1002/pmh.1434

- Weekers, L. C., Sellbom, M., Hutsebaut, J., Simonsen, S., & Bach, B. (2023). Normative data for the LPFS‐BF 2.0 derived from the Danish general population and relationship with psychosocial impairment. Personality and Mental Health, 17(2), 157-164. https://doi.org/10.1002/pmh.1570

Line 79-92. I think it could be emphasized more that the LPFS-SR is the only self-report instrument that takes both the vertical and horizontal structure of the LPFS into account.

Accomplished. Please see lines 82-88.

Line 93-111. I had troubles following along with the aims of the study because this section is merely a combination of research aims, methodological issues such as sample descriptions, the rationale for the study, and empirical findings. I hope the authors will be able to formulate the research questions more clearly. Also, what are “the original data”?

We have reformulated the research aims and hope that they are now more easily understandable. Please see lines 121-141.

Line 90-92. Do the authors suggest that maladaptive personality functioning is distinct from maladaptive personality content?

From a conceptual point of view, Criterion A relates to core impairment in self and interpersonal function (i.e., personality pathology severity) which is complemented by Criterion B’s description of the unique constellation of the pathological personality traits manifested (i.e., personality pathology style). However, from an empirical point of view, an overlap between Criterion A and Criterion B has been consistently documented (Morey, 2022). 

Please, see lines 90-101. 

- Morey, L. C., McCredie, M. N., Bender, D. S., & Skodol, A. E. (2022). Criterion A: Level of personality functioning in the alternative DSM–5 model for personality disorders. Personality Disorders: Theory, Research, and Treatment, 13(4), 305. https://doi.org/10.1037/per0000551

Line 106-111. The authors seem to suggest that there is no doubt that the LPFS-SR reflects an unidimensional factor whereas Sleep and colleagues are totally wrong. These formulations poorly reflect the discussion between Morey and Sleep about the dimensionality of the LPFS-SR. I think it would be wise to explain this discussion in more detail since it provides the main rationale for the current study. I also recommend to move this discussion to the introduction.

We have reformulated the paragraphs that refer to the debate around the unidimensionality of the LPFS-SR and have moved them to the introduction. Please see lines 102-118.

Line 148-164: I could not find any information about personality disorder assessment. How were PD diagnoses and symptom diagnoses assessed? If no structured clinical interviews were used, this should be mentioned as a limitation in the discussion. It is well known that clinical assessment of PDs without making use of structured diagnostic interviews, has poor diagnostic reliability. Thus, the lack of differences between the PD sample and non-PD sample could as well be due to poor assessment procedures.

In this sample collection, we did not use structured clinical interviews, however the assignment of diagnoses was a judicious process and followed a diagnosis-related group (DRG) system (Fetter et al., 1980).

In each affiliated mental health institution, a clinician coordinated the sampling procedures and selected the participants with the respective diagnoses contained in our study from the clinical databases of their institution, or from whom they had been referred. In general, our collected clinical samples relied on the direct clinical evaluation of several psychiatrists, whose diagnosis had been previously discussed and agreed upon by a clinical team. It should be noted that each diagnosis is the result of a medical psychiatric evaluation, held on the clinical records and conducted by at least three different clinicians: the assistant psychiatrist; the coding doctor, responsible for the respective Diagnosis-Related Groups (DRG) (Fetter et al., 1980); and the collaborating researcher.

The fact that we did not use a structured clinical interview to assess each diagnosis, which we agree may have hampered the diagnostic reliability, has been added as a limitation of this study. However, it should be noted that the LPFS-SR total score was able to discriminate between the PD and the OD sample (AUC = .63; p = .027).

- Fetter, R. B., Shin, Y., Freeman, J. L., Averill, R. F., & Thompson, J. D. (1980). Case mix definition by diagnosis-related groups. Medical care, 18(2), 1-53. http://www.jstor.org/stable/3764138

Overall, I think the manuscript could benefit from some more text-editing to make it more reader-friendly at some places. For instance, regarding the explanation of the translation process on line 156-164, the authors could write “obtained” instead of “requested” and “translator” instead of “author of the translation”. Moreover, it is not clear whether the back-translation was done by one person (a native English speaker who also was a professional translator) or two persons (a native English speaker and a professional translator).

We would like to thank Reviewer #2's suggestions regarding the explanation of the translation process which makes the text clearer. Please see lines 198-202.

To make the manuscript more reader-friendly, the text-editing of all the manuscript was performed by a native English speaker who is a professional translator. 

Line 182-188: I am not acquainted with the Factor software and have never seen other papers using this program. This might be due to lack of expertise on my part but it could also be due to the infrequent use of this program in the PD field. Therefore, I think it would be appropriate to give a more extensive explanation of this program in the Methods chapter. I would also recommend to restrict the explanation of the statistical procedures to the “Data analysis” section, i.e., move the explanation that “CFA and EFA are congregated” to the Methods chapter. It should be noted that, since I am not a psychometrician and do not have any background knowledge about the Factor package, my ability to assess the adequacy of the statistical analyses is limited.

We have tried to comply with Reviewer #2's request to explain the advantages of the Factor software, however we were unable to find a suitable part of the manuscript other than the data analysis section to include this information. Please see lines 205-212. 

Factor analysis is one of the multivariate statistical methodologies most used in Psychology, both for instrument validation and for reducing data dimensionality. Two approaches to factor analysis, namely Exploratory Factor Analysis (EFA) and Confirmatory Factor Analysis (CFA), are permanently confronted while their followers stand in opposition, wielding a huge set of arguments. Both aim to reproduce the relationships observed in a set of items and differ fundamentally in the nature of their specifications and a priori restrictions imposed by the model. An EFA is a data-based approach model, that is, no specifications are placed on either the number of factors or the dependency relationships between the items and factors. In the CFA model, the number of factors is specified a priori as well as the relational structure between the items and factors, namely the imposition in the CFA that each item can only be loaded onto a single factor. 

The FACTOR software freely obtained from Rovira i Virgili University, is one of the most complete statistical programs for conducting an EFA, given that it congregates both CFA and EFA indexes. It includes the most current criteria for fundamental decisions: the correlation matrix to be used, the method to be adopted for extracting common factors, the number of factors to be retained and the rotation method. Additionally, this software provides some goodness-of-fit indexes and the corresponding confidence intervals are based on bootstrap techniques, which are unusual in EFA algorithms.

- Ferrando, P. J., & Lorenzo Seva, U. (2017). Program FACTOR at 10: Origins, development and future directions. Psicothema, 29(2), 236-241.

- Rogers, P. (2022). Best practices for your exploratory factor analysis: A factor tutorial. Revista de Administração Contemporânea, 26(06). https://doi.org/10.1590/1982-7849rac2022210085.en

Line 321-343: In scientific papers, it is common to provide several explanations for unexpected findings. I have already mentioned the possibility of poor diagnostic reliability. Could there be other explanations for the finding that the LPFS-SR could not make a distinction between the two clinical samples? Such a discussion is more interesting and informative than just a repetition of the results and a conclusion that more research is needed.

Please see lines 404-415 and 428-432.

However, it should be noted that the LPFS-SR total score was able to discriminate between the PD and the OD sample (AUC = .63; p = .027).

Line 340-343: the information in the section is not correct. Self pathology and interpersonal problems are the core features of PD, by definition.

We believe there has been a misunderstanding. We are not saying that self-pathology and interpersonal problems are not the core features of PD, but rather that the LPFS also captures impairments in self and interpersonal functioning underlying other psychiatric diagnoses. In other words, what we mean to say is that in a depressive disorder, for instance, there are also impairments in self and interpersonal functioning. Criterion A relates to personality and personality shapes psychopathological manifestations.

Perhaps the following formulation is clearer?

“Moreover, it should be noted that the LPFS model not only seeks to describe the core features of PD, but may also characterize impairments in self and interpersonal functioning underlying many forms of psychopathology.” (Lines 409-411)

- Bender, D. S. (2019). The P-factor and what it means to be human: commentary on criterion A of the AMPD in HiTOP. Journal of personality assessment, 101(4), 356-359. https://doi.org /10.1080/00223891.2018.1492928

Line 344-355. This is an important section, which needs thorough revision. First, I couldn’t find back the proposed threshold values in the paper of Morey et al. (Morey, 2017). I neither could find any information about the interpretative criterion in this paper. Could the authors give some more explanation of these “threshold values” and “interpretative criterion”? I also had difficulties understanding how cultural factors could have increased the threshold in the Iranian sample and why we should address interpretability of the LPFS-SR scores. Could this be explained more clearly? To me, it seems that sample characteristics are more important than possible cultural factors. After all, the sample of Hemmati et al. consisted of inpatients, most of whom were diagnosed with either borderline PD or antisocial PD. Moreover, I did not manage to understand the last sentence of this paragraph. Could the authors’ point be explained more clearly?

We have followed the suggestion of Reviewer # 1 and reported only the ROC contrasts between the PD sample and the community sample and the PD sample and the other clinical sample.

We agree that these contrasts are the most relevant for the validity of a measure that pertains to assess features relevant to a PD diagnosis. 

Thus, we have decided to exclude from the text the reference to the optimal cut-off to discriminate between the clinical and non-clinical population. The decision to mention it derived from Hemmati et al.'s study, however since this was also questioned by Reviewer #2, the text has been reformulated.

Please see lines: 227-229; 314-318; 401-415.

Line 359: It is a common error to interpret a large Chronbach’s alpha value as proof for unidimensionality. The authors could easily avoid this error by focusing more on the results of the factor analyses.

This methodological option was taken in the interest of comparing this study's results with Hemmati et al.'s study (2020) but since this was called into question, the text has been reformulated. Please see lines 362-384.

Line 365-381: The section about limitations should have a stronger on the very limitations rather than repeating the results. How could these imitations have influenced the results? For instance, what kind of errors could be associated with the small size of the PD sample? And how could comorbidity have influenced the results? (Is this really a limitation? A clinical sample without any comorbidity would be a rarity and not very representative.) If it is correct that no structured diagnostic instruments were used, this should also be mentioned as a limitation. Finally, I am not sure if it is wise to write “Criterion A’s opponents”. In my view, this formulation has a rather negative connotation in this context. I assume that the authors are referring to Sleep & colleagues. Sure, these authors are critical to the view that the LPFS is a unidimensional construct but I really cannot see that they are opposing the A criterion.

Regarding the sentence related to “Criterion A’s opponents”, we have changed it as we are not interested in Manichaean views. Our one and only main interest is to share the data we have obtained which supports the unidimensionality of the LPFS-SR. Please see lines 416-423.

However, we are of the opinion that Sleep et al. (2022) really do oppose Criterion A. Please consider the following reference that led us to conclude this and share part of its conclusions:

“As demonstrated here, the LPFS does not discriminate PD pathology from other forms of psychopathology, does not account for patterns of comorbidity among the PDs, shows problematic levels of overlap with both disordered and nondisordered personality traits, fails to provide incremental validity over these same traits, and has an unstable internal structure. In short, it does not do what it is supposed to do. Until a better system comes along, the traits themselves carry the most useful information.”

- Sleep, C. E., & Lynam, D. R. (2022). The problems with Criterion A: A comment on Morey et al.(2022). Personality Disorders: Theory, Research, and Treatment, 13(4), 325–327. https://doi.org/10.1037/per0000585

---

## [Decision Letter · Decision Letter 1]

2 Jan 2024

PONE-D-23-22309R1The Portuguese version of the self-report form of the DSM-5 Level of Personality Functioning Scale (LPFS-SR) in a community and clinical samplePLOS ONE

Dear Dr. Pires,

Thank you for submitting your manuscript to PLOS ONE. After careful consideration, we feel that it has merit but does not fully meet PLOS ONE’s publication criteria as it currently stands. Therefore, we invite you to submit a revised version of the manuscript that addresses the points raised during the review process.

**ACADEMIC EDITOR: **Thank you for the review you made in this paper, taking into account the comments sent. The result is very close to a version acceptable for publication. However, reviewers made some more minor comments that I think will improve your paper and strenghen the discussion.

Please read carefully and incorporate them in your manuscript.============================== Please submit your revised manuscript by Feb 16 2024 11:59PM. If you will need more time than this to complete your revisions, please reply to this message or contact the journal office at plosone@plos.org. Please include the following items when submitting your revised manuscript:A rebuttal letter that responds to each point raised by the academic editor and reviewer(s). You should upload this letter as a separate file labeled 'Response to Reviewers'.A marked-up copy of your manuscript that highlights changes made to the original version. You should upload this as a separate file labeled 'Revised Manuscript with Track Changes'.An unmarked version of your revised paper without tracked changes. You should upload this as a separate file labeled 'Manuscript'.If applicable, we recommend that you deposit your laboratory protocols in protocols.io to enhance the reproducibility of your results. Protocols.io assigns your protocol its own identifier (DOI) so that it can be cited independently in the future. For instructions see: https://journals.plos.org/plosone/s/submission-guidelines#loc-laboratory-protocols. Additionally, PLOS ONE offers an option for publishing peer-reviewed Lab Protocol articles, which describe protocols hosted on protocols.io. Read more information on sharing protocols at https://plos.org/protocols?utm_medium=editorial-email&utm_source=authorletters&utm_campaign=protocols.

We look forward to receiving your revised manuscript.

Kind regards,

Paulo Santos, PhD

Academic Editor

PLOS ONE

Journal Requirements:

Additional Editor Comments:

This paper is almost acceptable for publishing. However, reviewers introduced several minor comments that would improve the final version and strengthen the discussion.

Reviewers' comments:

Reviewer's Responses to Questions

**Comments to the Author**

1. If the authors have adequately addressed your comments raised in a previous round of review and you feel that this manuscript is now acceptable for publication, you may indicate that here to bypass the “Comments to the Author” section, enter your conflict of interest statement in the “Confidential to Editor” section, and submit your "Accept" recommendation.

Reviewer #3: All comments have been addressed

Reviewer #4: All comments have been addressed

Reviewer #5: (No Response)

2. Is the manuscript technically sound, and do the data support the conclusions?

Reviewer #3: Yes

Reviewer #4: Yes

Reviewer #5: Yes

3. Has the statistical analysis been performed appropriately and rigorously? 

Reviewer #3: Yes

Reviewer #4: No

Reviewer #5: Yes

4. Have the authors made all data underlying the findings in their manuscript fully available?

Reviewer #3: Yes

Reviewer #4: Yes

Reviewer #5: Yes

5. Is the manuscript presented in an intelligible fashion and written in standard English?

Reviewer #3: Yes

Reviewer #4: Yes

Reviewer #5: Yes

6. Review Comments to the Author

Reviewer #3: After reading the revised manuscript, I think that the authors addressed all the reviewers´ concerns.

Reviewer #4: Dear Editor,

Thank you for giving me the opportunity to share my thoughts on the manuscript entitled "The Portuguese version of the self-report form of the DSM-5 Level of Personality Functioning Scale (LPFS-SR) in a community and clinical sample". The paper provides valuable results that may contribute to the cross-cultural generalizability of current personality models. In my opinion, the authors have carefully answered all the comments and suggestions of the journal reviewers in the previous stage. However, I have some more suggestions that I think will help provide a better work.

• What was the criteria of the authors to determine the LPFS-SR cutoff scores? For example, Youden index or Gini index? The following reference may be a good pattern for the authors:

- Komasi S, Rezaei F, Hemmati A, Nazari A, Nasiri Y, Faridmarandi B, Zakiei A, Saeidi M, Hopwood CJ. Clinical cut scores for the Persian version of the personality inventory for DSM-5. J Clin Psychol. 2023. https://doi.org/10.1002/jclp.23614

• According to the cut-off scores of the scale, the authors can report the true positive rate (TPR) and the true negative rate (TNR).

• What is the similarity between the extracted factors with international studies? I recommend reporting Tucker’s congruence coefficients.

• Although examining convergent validity was probably not one of the goals of the study, if data are available, reporting correlations between LPFS-SR and other dimensional measures of personality would provide readers with additional information.

Reviewer #5: Dear authors,

Thank you for the opportunity to review the manuscript "The Portuguese version of the self-report form of the DSM-5 Level of Personality Functioning Scale (LPFS-SR) in a community and clinical sample".

It is an interesting study and the manuscript is very clear and well written. I believe that the reviewers' comments helped to improve the understanding of the content of this study and all the suggestions from the same reviewers were responded to in an assertive manner.

My only additional suggestion is that the authors mention the limitations of the study is that the validation of the Portuguese version of the LPFS-SR is not yet complete. This is due to the absence of scale reliability tests. According to COSMIN, in addition to internal consistency, reliability is an umbrella clinimetric property that encompasses reliability (the ability of the instrument to detect similar responses in repeated applications on the same patient, if he/she is clinically stable) and the error of measurement (size of the error systematically present during the application of the instrument). Along with criterion and discriminant validity, reliability would ensure the complete accuracy of the scale for use in clinical practice and scientific research. However, the reliability test depends on a test and retest design, and this was probably the reason for its failure.

7. PLOS authors have the option to publish the peer review history of their article (what does this mean?). If published, this will include your full peer review and any attached files.

Reviewer #3: No

Reviewer #4: **Yes: **I agree that my name will be published as a reviewer of the article.

Reviewer #5: **Yes: **Adriana C Lunardi

---

## [Author Response · Author response to Decision Letter 1]

14 Feb 2024

The Portuguese version of the self-report form of the DSM-5 Level of Personality Functioning Scale (LPFS-SR) in a community and clinical sample [PONE-D-23-22309R1] 

RESPONSE TO REVIEWERS

We would like to thank Reviewers 3, 4 and 5 for recognizing and appreciating the amendments made, as well as for encouraging us to continue improving the paper. 

REVIEWER 4

In the following lines we respond to Reviewer 4’s comments and suggestions. Our comments and inserted text are in blue, and follow the order in which they were raised. 

1. What was the criteria of the authors to determine the LPFS-SR cutoff scores? For example, Youden index or Gini index? 

ROC curve analyses were used to examine the capacity of the LPFS-SR total score in discriminating between the samples. Optimal cutoff scores were determined to maximize the sum of sensitivity and specificity, ensuring the fewest incorrect decisions under conditions of equal a priori probabilities.

The Youden index is calculated as Sensitivity + Specificity - 1, and the Gini coefficient is given by 2 × AUC − 1.

2. According to the cut-off scores of the scale, the authors can report the true positive rate (TPR) and the true negative rate (TNR).

In our manuscript, we reported the sensitivity (true positive rate) and the specificity (true negative rate).

3. What is the similarity between the extracted factors with international studies? I recommend reporting Tucker’s congruence coefficients.

Tucker’s congruence coefficient is an index of the similarity between factor results (Lorenzo-Seva et al., 2006). The most popular tool for such comparisons was initially proposed by Burt (1948) and later gained popularity as Tucker’s congruence coefficient (Tucker, 1951). Its values range between -1 and +1.

Tucker’s congruence coefficient is defined as φ_C= (∑▒〖x_i y_i 〗)/√(∑▒〖x_i^2 ∑▒y_i^2 〗^ ) where xi and yi are the loadings of variable i on factor X and Y, respectively, i=1, …, n.

As suggested, we determined the Tucker’s congruence coefficient between the factor results of the Portuguese community sample, the Portuguese clinical sample, and the item factor loadings of Bliton et al. (2022). Unfortunately, we couldn't find other international factor results to compute the Tucker coefficient. Please see lines 235-238 and 382-391.

- Burt, C. (1948). The factorial study of temperament traits. British Journal of Psychology, Statistical Section, 1, 178-203.

- Lorenzo-Seva, U., & Ten Berge, J. M. (2006). Tucker's congruence coefficient as a meaningful index of factor similarity. Methodology, 2(2), 57-64.

- Tucker, L. R. (1951). A method for synthesis of factor analysis studies. Personnel Research Section Report No.984, Washington D.C.: Department of the Army.

4. Although examining convergent validity was probably not one of the goals of the study, if data are available, reporting correlations between LPFS-SR and other dimensional measures of personality would provide readers with additional information.

Data examining the LPFS-SR convergent validity with other dimensional measures of personality is already published. Please see the following reference for more details:

- Pires, R., Henriques-Calado, J., Sousa Ferreira, A., Gama Marques, J., Ribeiro Moreira, A., Barata, B. C., Paulino, M., & Gonçalves, B. (2023). Bridging the ICD11 and the DSM-5 personality disorders classification systems: The role of the PID5BF + M. Frontiers in Psychiatry, 14, 1004895. https://doi.org/ 10.3389/fpsyt.2023.1004895 

REVIEWER 5

1. My only additional suggestion is that the authors mention the limitations of the study is that the validation of the Portuguese version of the LPFS-SR is not yet complete. This is due to the absence of scale reliability tests. According to COSMIN, in addition to internal consistency, reliability is an umbrella clinimetric property that encompasses reliability (the ability of the instrument to detect similar responses in repeated applications on the same patient, if he/she is clinically stable) and the error of measurement (size of the error systematically present during the application of the instrument). Along with criterion and discriminant validity, reliability would ensure the complete accuracy of the scale for use in clinical practice and scientific research. However, the reliability test depends on a test and retest design, and this was probably the reason for its failure.

We have followed Reviewer 5's suggestion, which has been implemented in the following lines: 432-436.

---

## [Editor Report · Decision Letter 2]

5 Mar 2024

The Portuguese version of the self-report form of the DSM-5 Level of Personality Functioning Scale (LPFS-SR) in a community and clinical sample

PONE-D-23-22309R2

Dear Dr. Pires,

We’re pleased to inform you that your manuscript has been judged scientifically suitable for publication and will be formally accepted for publication once it meets all outstanding technical requirements.

Kind regards,

Paulo Alexandre Azevedo Pereira Santos, PhD

Academic Editor

PLOS ONE
---

## [Editor Report · Acceptance letter]

23 Mar 2024

PONE-D-23-22309R2 

PLOS ONE

Dear Dr. Pires, 

I'm pleased to inform you that your manuscript has been deemed suitable for publication in PLOS ONE. Congratulations! Your manuscript is now being handed over to our production team.

Kind regards, 

on behalf of

Professor Paulo Alexandre Azevedo Pereira Santos 

Academic Editor

PLOS ONE